# A Non-Canonical Pathway Induced by Externally Applied Virus-Specific dsRNA in Potato Plants

**DOI:** 10.3390/ijms242115769

**Published:** 2023-10-30

**Authors:** Viktoriya O. Samarskaya, Nadezhda Spechenkova, Irina Ilina, Tatiana P. Suprunova, Natalia O. Kalinina, Andrew J. Love, Michael E. Taliansky

**Affiliations:** 1Shemyakin-Ovchinnikov Institute of Bioorganic Chemistry, Russian Academy of Sciences, 117997 Moscow, Russia; viktoriya.samarskaya2012@yandex.ru (V.O.S.); rysalka47@gmail.com (N.S.); irinailina.bio@gmail.com (I.I.); kalinina@belozersky.msu.ru (N.O.K.); 2Doka-Gene Technologies Ltd., Rogachevo, 141880 Moscow, Russia; suprunova@gmail.com; 3Belozersky Institute of Physico-Chemical Biology, Lomonosov Moscow State University, 119991 Moscow, Russia; 4The James Hutton Institute, Invergowrie, Dundee DD2 5DA, UK; andrew.love@hutton.ac.uk

**Keywords:** potato virus Y, dsRNA-mediated virus resistance, small RNAs, exogenously applied dsRNA

## Abstract

The external application of double-stranded RNA (dsRNA) has recently been developed as a non-transgenic approach for crop protection against pests and pathogens. This novel and emerging approach has come to prominence due to its safety and environmental benefits. It is generally assumed that the mechanism of dsRNA-mediated antivirus RNA silencing is similar to that of natural RNA interference (RNAi)-based defence against RNA-containing viruses. There is, however, no direct evidence to support this idea. Here, we provide data on the high-throughput sequencing (HTS) analysis of small non-coding RNAs (sRNA) as hallmarks of RNAi induced by infection with the RNA-containing potato virus Y (PVY) and also by exogenous application of dsRNA which corresponds to a fragment of the PVY genome. Intriguingly, in contrast to PVY-induced production of discrete 21 and 22 nt sRNA species, the externally administered PVY dsRNA fragment led to generation of a non-canonical pool of sRNAs, which were present as ladders of ~18–30 nt in length; suggestive of an unexpected sRNA biogenesis pathway. Interestingly, these non-canonical sRNAs are unable to move systemically and also do not induce transitive amplification. These findings may have significant implications for further developments in dsRNA-mediated crop protection.

## 1. Introduction

Plant viruses are devastating plant pathogens that severely affect crop yield and quality and significantly curtail food production worldwide [1]. To counteract viral infections, plants have evolved various passive or active defence mechanisms, including RNA-interference (RNAi or RNA silencing), which can act as a central antiviral pathway [2,3,4]. RNAi is a complex family of inter-related mechanisms which controls plant gene expression and also influences virus pathogenicity via sequence-specific gene regulation by small non-coding RNAs (sRNAs), such as microRNAs (miRNAs), or small interfering RNAs (siRNAs) [5,6]. miRNAs are important in host plant regulation of gene expression and are encoded by endogenous MIRNA (MIR) genes which are transcribed by DNA-dependent RNA polymerase II (Pol II) to produce primary miRNAs (pri-miRNAs) that can fold into hairpin (hp) miRNA precursors [4,7]. These miRNAs function as targeting molecules which can control, in a complementary sequence-specific manner, the posttranscriptional regulation of diverse gene expressions, including those governing plant growth, differentiation, development, stress responses and immunity. In contrast to miRNAs, which are produced endogenously, siRNA can be exogenous or endogenous in origin. siRNAs are derived from dsRNA and induce the silencing machinery to target complementary DNA or RNA for transcriptional (TGS; i.e., nucleic acid methylation) or posttranscriptional (PTGS; i.e., translational repression or degradation of specific RNAs) gene silencing, respectively [5,6,8]. In the context of RNA viruses, dsRNA molecules are formed during virus replication, which may become derivatised into siRNA. 

Generally, these natural RNAi pathways involve (i) the perception and cleavage of hpRNAs or dsRNAs into primary miRNAs or siRNAs, respectively, by RNase III-like enzymes referred to as Dicer-like (DCL) [5,6], (ii) production of a second wave of sRNAs, known as secondary siRNAs [9]; generation of these molecules involves RNA-dependent RNA polymerase (RdRP) protein family members, which use the target single-stranded RNA (ssRNA) as a template for the synthesis of new dsRNA molecules, (iii) the loading of siRNA/miRNAs into an RNA-Induced Silencing Complex (RISC) [10,11] which facilitates (iv) the complementary base pairing between siRNA/miRNA with a targeted RNA or DNA sequence, and the nucleolytic cleavage or repression of the target via the activity of an Argonaute (AGO) protein [10]. Plants encode multiple highly specialised DCL, RdRP and AGO proteins which operate in a coordinated manner and in concert with certain other plant proteins to ensure selective segregation of siRNAs into distinct RNAi pathways which predetermine their biological function. Arabidopsis, for example, encodes four DCL and ten AGO genes directly involved in the RNA-silencing pathways [10,12]. Of the six RdRP genes identified in Arabidopsis, three of them (RdRP1, RdRP2 and RdRP6) were shown to play roles in RNAi [12].

Extensive genetic studies clearly demonstrated that each DCL, RdRP or AGO protein participates in a specific RNA-silencing pathway, with some redundancy [4,12]. As a result, functional diversification of the RNAi pathways may explain the plethora of endogenous and exogenous targets (including viral sequences) which may be processed. 

DCL1 produces miRNAs, usually 21 nucleotides (nt) in length, from endogenous hpRNA precursors [12]. DCL3 recognises relatively short dsRNA precursors of ~30–50 nt, which were synthesized by NUCLEAR RNA POLYMERASE IV (Pol IV) and RdRP2 and then processes them into 24 nt siRNA duplexes [13]. In contrast, DCL2 and DCL4 cleave long dsRNAs (longer than 50 nts) into 22 and 21 nt siRNAs, respectively [12,13]. The majority of sRNAs, including miRNAs and siRNAs, are produced as duplexes from precursors by DCL proteins and are subsequently loaded into AGO proteins. To achieve their diverse regulatory functions, sRNAs are sorted into the correct AGO protein based on their length and 5′ nucleotide identity [14]. For example, AGO1 and AGO2 preferentially load 21/22 nt sRNAs bearing a uridine (U) and adenine (A) nucleotide at the 5′ end, respectively. AGO5 binds all three size classes (21, 22 and 24 nt) of sRNAs with a bias towards a 5′ terminal cytosine (C). Plant 24-nt siRNAs induce RNA-directed DNA methylation (RdDM) by associating with AGO4 or AGO6 which have a strong preference to 5′A-containing siRNAs. Other additional factors that may determine the specificity of sRNA sorting into AGOs can include subcellular compartmentalization of RNA silencing machinery and signals originating from the upstream sRNA biogenesis machinery [11,14]. 

Before associating with the correct AGO, an siRNA duplex consists of two RNA strands: a guide (or antisense) strand and a passenger (or sense) strand. During the interaction with AGO, the passenger strand, which is recognized by the preferred 5′ nucleotide at the thermodynamically less stable siRNA terminus, is ejected from the complex and becomes degraded, while the other strand remains bound to AGO. This constitutes the mature RISC, which is directed to silence target RNAs which have a complementary sequence to the attached strand [15]. 

Another important player in RNAi pathways are RdRPs, as they are implicated in a robust RNAi enforcing mechanism referred to as transitivity [12,16,17]. Transitivity in plants relies on RdRP-mediated dsRNA synthesis using siRNA-targeted RNA. This secondary dsRNA is subsequently cleaved into secondary siRNAs by DCL endonucleases. As with primary siRNAs, secondary siRNAs are also loaded into AGO proteins to form RISCs, which reinforces cleavage of the target RNA. Thus, as a result of transitivity, the silencing signal is significantly expanded to additional sequences of the transcript since RdRP transcription can proceed for up to 700 nt from the start towards the 5′ end [17,18]. It should also be noted that, in plants, RNA silencing is a non-cell-autonomous event, initiated in single cells with eventual expansion from cell to cell or even systemically throughout the whole plant. Intercellular and long-distance movement of non-cell autonomous RNA silencing involves sRNAs, and the capacity to trigger transitivity might be an important factor in determining sRNAs as functional mobile RNAi signals [9].

In the context of RNA viruses, antiviral RNAi is initially triggered by dsRNA molecules produced during replication of the viral genome [4,12]. These replicated intermediate dsRNA molecules are recognised by DCL4 or DCL2 and cleaved into 21 and 22 nt siRNAs, respectively. Viral ssRNAs are also converted into dsRNAs by host RdRPs which act as new precursors of secondary siRNAs, and this is critical for effective and amplified RNAi responses to virus infection. Secondary siRNAs significantly enforce defence against viruses [4,12]. DNA viruses may be transcribed convergently on opposite DNA strands so that overlapping transcripts may base-pair to form dsRNA. A likely scenario in RNA-directed silencing of DNA viruses is that this dsRNA can initially be recruited in the same RNAi pathway as RNA viruses, and then, if the RNA silencing is weak, there could be a transition to DNA methylation and resultant transcriptional inhibition via DCL3-directed RdDM [4]. 

Viruses encode suppressors of RNA silencing (VSRs) which counteract RNA silencing by preventing DCL’s access to sRNA, or by sequestering the sRNA away from an AGO, or via targeted proteolytic degradation of RNAi pathway machinery components, or through their effects on RNA-silencing signal mobility [4,19].

Great progress in RNAi research and its role in antiviral immunity has opened new opportunities for deploying increasingly powerful RNAi-based technologies in crop improvement for viral resistance. Generation of virus-derived dsRNA is a general feature when successful resistance is achieved in plants. Many approaches have been developed for engineering virus-resistant transgenic plants, mostly based on different precursor RNA for siRNA production, including sense/antisense RNA, hpRNA and artificial miRNA precursors (host-induced gene silencing, HIGS) [20]. However, the approval of RNAi-based transgenic plants has always been challenging due to public concern over the production of genetically modified crops (GMOs). An increasing number of applications have emerged involving the exogenous application of dsRNA (such as so called “spray-induced gene silencing”, SIGS) which provide environmentally friendly options for crop protection [21,22,23]. Thus far, such exogenous technologies have been successfully applied to target over 10 different economically important plant viruses in more than 10 plant species [24]. Interestingly, spraying of dsRNA molecules operates to not only protect plants from viruses but also other pests and pathogens, including, for example, oomycetes such as *Phytophthora infestans* [25].

It is generally assumed that the mechanism of dsRNA-mediated antivirus RNA silencing is quite similar (or even identical) to that of natural RNAi-based defence against RNA-containing viruses [24,26,27,28,29,30,31,32,33]. There is, however, no direct evidence to support this idea. Here we provide data on the comparative analysis of sRNA (as hallmarks of RNAi) induced by infection with the RNA-containingPVY and also by exogenous application of dsRNA which corresponds to a fragment of the PVY genome. Intriguingly, in contrast to PVY-induced production of discrete 21 and 22 nt siRNA species, the externally administered PVY dsRNA fragment led to generation of a non-canonical pool of sRNAs, which were present as ladders of ~18–30 nt in length; suggestive of an unexpected sRNA biogenesis pathway. These findings may have significant implications for further developments in dsRNA-mediated crop protection.

## 2. Results

### 2.1. Impact of PVY-Specific dsRNA on PVY Accumulation in Potato Plants

To study a mechanism underpinning dsRNA-mediated defence responses of potato plants against PVY-NTN, *S. tuberosum* (potato) plants cv. Indigo were treated with buffer only or bacterially produced PVY-specific dsRNA (dsRNApvy) molecules homologous to the conservative fragment of the PVY replicase gene with a length of 500 bp (nts 7739–8238; Gene Bank accession number OR545670). This (*Nib*) genetic region of PVY is highly conserved; our preliminary bioinformatics analysis predicted that this region is capable of producing the maximum siRNAs, which makes it a good choice as a favorable target. PVY was inoculated on to these plants 24 h later. Application of dsRNApvy caused efficient inhibition of viral propagation in systemically infected leaves of the treated plants until at least 14 days post-infection (dpi) compared with the control treatments (plants buffer-treated rather than sprayed with dsRNA or plants treated with non-target potato virus S-specific dsRNA (dsRNApvs)) (Figure 1). However, with time the PVY dsRNA-mediated virus suppression decreased by 14 dpi and eventually disappeared (by 19 dpi) allowing the virus accumulation to reach similar levels to that of the control (Figure 1). In contrast, in control plants treated with either buffer or non-specific dsRNApvs, no protection against PVY was observed (Figure 1). These results are consistent with numerous reports which suggest that externally applied dsRNAs act in a sequence-specific manner similar to RNAi antiviral defence (see for review, [24].)

Interestingly, we previously showed that the antivirus effect of dsRNA could be prolonged by repetitive dsRNA treatments on a weekly basis [33]. However, in these experiments we deliberately did not use repetitive treatments in order to prevent possible data misinterpretation caused by imposition of consecutive dsRNA applications.

### 2.2. Persistence and Systemic Movement of PVY RNA-Targeting dsRNA

There are some contradictory results about the ability of foliar-applied dsRNA to spread systemically in treated plants. Several previous reports described the systemic movement of dsRNA in tobacco and tomato plants [27,29,30,32]. In contrast, works by Tenllado and Díaz-Ruíz [34] and Rego-Machado et al. [31] did not reveal the presence of dsRNA in the non-treated leaves when dsRNA was exogenously applied to N. tabacum and tomato plants. In order to examine the persistence and the systemic movement of dsRNApvy in potato plants we used high-throughput sequencing (HTS) Illumina technology to evaluate the presence of dsRNApvy in non-treated leaves of plants or those exogenously treated with dsRNA. Sixteen paired-end RNA-Seq libraries were generated from four biological replicates of PVY inoculated or dsRNApvy-treated leaves and also upper untreated and uninoculated leaves harvested therefrom at seven days post treatment (dpt). A summary of the sequencing statistics and mapping is presented in Appendix A.

Our analysis of sequence coverage maps revealed that, as could be expected, reads in all libraries generated from both inoculated and uninoculated leaves of PVY-infected plants were relatively evenly distributed across the whole PVY genome. With regard to dsRNApvy treatment only, all samples from dsRNApvy-treated leaves (at 7 dpt) produced RNA reads that exclusively matched the PVY genome region, which was used for dsRNA design and construction (nts 7739–8238) (Figure 2A). Moreover, a number of reads induced by dsRNA in this dsRNApvy target region were markedly higher than those induced by PVY infection, suggesting the capacity of the dsRNApvy to penetrate leaf tissues, although its exact intra-specific leaf localisation remains unclear.

Concerning systemic movement of dsRNApvy, its sequences were detected in untreated leaves but only in two of the four dsRNApvy-treated plants. Moreover, the number of such reads in untreated leaves was significantly lower than in treated leaves (Figure 2B). It is worth noting that “non-treated” leaves used for analysis only emerged sometime after the treatment, and these were collected with maximal sterile precautions. Despite this we cannot completely exclude the possibility that contamination of untreated leaves with some amounts of dsRNA may have taken place. However, it is more likely the case that the presence of dsRNA in those leaves would be due to *in planta* physiological movement, which is consistent with the previous reports mentioned above (see for review, [24]). Such movement of dsRNA may have a limited capacity within potato plants, but anyway, the majority of them remain in treated leaves.

### 2.3. Analysis of sRNA Molecules

The sRNA sequencing by HTS was performed to evaluate the sRNA populations generated from the exogenously applied dsRNApvy and the virus. The following treatments were analysed: (i) PVY, (ii) dsRNApvy and (iii) PVY + dsRNApvy in both treated (inoculated) and untreated (uninoculated systemically infected) leaves. Twenty-four single-end RNA-Seq libraries were generated from four biological replicates of mock-inoculated and PVY-infected plants at 7 dpt. A summary of the sequencing statistics and mapping is present in Appendix A.

The sRNA read length analysis demonstrated that both inoculated (Figure 3A) and upper uninoculated (Figure 3B) leaves of PVY infected plants contained canonical virus-specific siRNA species of 21 nt and 22 nt in size, with 21 nt species being predominant. This is entirely consistent with the general size distribution of the sRNA species induced by other RNA-containing plant viruses [4,12]. siRNAs of 21 nt and 22 nt in size are normally produced by DCL4 and DCL2, respectively, and are then uploaded into AGO1 or AGO2 to form RISC to mediate the RNA slicing or translational inhibition of RNA viruses [11,12]. 

In contrast, in the dsRNA-treated leaves of PVY-uninfected plants (at 7 dpt) sRNAs were present as a ladder of 18–30 nt in length, with longer fragments having progressively decreased read counts (Figure 3C). This suggests that the detected sRNAs were unlikely to have been produced by DCLs, as DCLs are known to generate sRNAs of discrete sizes [35]. Nevertheless, all samples from these leaves produced RNA reads that exclusively matched the PVY genome segment, which was used for dsRNA design and construction, clearly implicating the sequence-specific origin of the non-canonical 18–30 nt population of sRNAs (Figure 3C,D and Figure 4). 

Interestingly, similar ladder-like size distributions of sRNA caused by external application of dsRNAs was described by Uslu et al. [36], Nityagovsky et al. [28], Tabein et al. [30] and Rego-Machado et al. [31]. However, results were contradictory: while Nityagovsky et al. [28] showed the silencing effect on two endogenic/transgenic plant genes including GFP, Uslu et al. [36] did not detect any effect on GFP. In two other works [30,31], the formation of ladder-like sRNA size distributions in *N. benthamiana* and tomato plants correlated with virus suppression. Reasons for these discrepancies are not clear. Our work also confirms that externally applied dsRNA induces the formation of ladder-like sRNA species and sequence-specific virus suppression in potato plants, *in toto* suggesting that that these two processes are mechanistically interlinked.

To examine whether non-canonical sRNA molecules can move systemically, we investigated the amounts and sizes of PVY-specific RNA reads obtained from upper non-treated leaves of plants whose bottom leaves were treated with externally applied dsRNApvy. sRNAs were found in untreated leaves at 7 dpt, but only in two of the four treated plants; indicating that dsRNA successfully moved systemically from treated leaves and accumulated in untreated leaves of those plants. In contrast, in dsRNApvy-treated plants deficient in dsRNA systemic movement, the accumulation of sRNAs was negligible in the untreated leaves.

These data suggest that dsRNA which move systemically can be cleaved in upper untreated leaves into non-canonical sRNAs in a manner similar to that in treated leaves. With regard to the ability of non-canonical sRNAs themselves to move systemically, we were not able to detect any signs of their effective mobility (if any). 

Further analyses were performed to investigate the polarity of sRNA. By aligning sRNAs to the PVY-NTN genome sequence, we found that the numbers of reads with sequences complementary to the virus genome (for the sake of simplicity, hereinafter referred to as antisense sRNAs; 64.9%) predominated over the number of opposite strand-specific reads (hereinafter referred to as sense sRNAs; 35.1%) for all sRNAs size classes (18–30 nt) in dsRNApvy-treated leaves of uninfected plants (Figure 3C and Figure 4). Interestingly, a similar trend was observed in the untreated leaves of those uninfected dsRNApvy treated plants; a higher number of anti-sense sRNA reads exceeded that of sense sRNA reads (Figure 3D). Virus-induced sRNAs detected in inoculated and uninoculated (systemically infected) leaves of PVY-infected plants were derived from both strands of the entire PVY genome but there was some bias towards the sense strand (Figure 3A,B and Figure 4). The asymmetry in strand polarity (with bias towards either sense or antisense siRNAs) is well documented for virus infections and is usually explained by the effect of virus-specific factors, such as viral ssRNA itself or VSRs of RNA silencing [37,38,39], as will be discussed in more detail below. However, why and how two dsRNA strands could be distinguished from each other by the plant and only one of them would be selected for a biased sRNA accumulation in the absence of the virus remains puzzling and challenging.

As mentioned above, in addition to sRNA size, its 5′-terminal nucleotide plays an important role in the selective loading of sRNAs into specific AGOs [10,11,40]. Therefore, to estimate the amount of sRNA potentially able to produce RISCs which may actively silence PVY, we examined the relative abundance of the four different 5′ nucleotide identities in the antisense sRNAs which may actually serve as a guide for their incorporation into active RISCs (Figure 5). In PVY infection, two major classes of antisense sRNAs (21 nt and 22 nt) were enriched in 5′U (~36–40%) and 5′A (~28–29%), followed by 5′C (~20–21%) in both inoculated and systemically infected leaves, suggesting their association with AGO1-, AGO2-, and AGO5-like proteins, respectively (Figure 5C,D), as predicted from the data obtained on model Arabidopsis plants [10,11]. 

In spite of the variable distribution of nucleotides at the 5′-ends of the non-canonical (18–30 nt) sRNAs species detected in dsRNApvy treated plants, they still have potential affinity (sufficient number of A, U or C) to the same repertoire of AGO proteins (AGO1, AGO2 and AGO5) as PVY-induced siRNAs (Figure 5; Appendix A). At the same time, it is unclear if any sRNAs found in dsRNApvy-treated plants, which are actually non-canonical sRNAs (with atypical size), are able to form RISCs with AGO proteins. 

### 2.4. Effect of PVY Infection on Biogenesis of Non-Canonical sRNA Produced by PVY

Non-canonical sRNAs produced by external dsRNApvy have a ladder-like size distribution (Figure 3C) and as such can be easily distinguished from canonical siRNAs produced during PVY infection (Figure 3A). Comparative analysis of the siRNA profile in dsRNApvy-treated leaves which were also infected with PVY clearly demonstrated the presence of a nearly whole spectrum of non-canonical (18–30 nt) sRNAs which matched the PVY genome region that was used for dsRNA construction (Figure 6A). In the case of virus infections, viral ssRNA may serve as a template which is primed by primary siRNAs for RdRP-mediated production of new dsRNAs which are further processed by DCL2 and DCL4 to produce secondary siRNAs [9] that degrade complementary mRNAs. Transitive RNA silencing can spread both upstream and downstream of the primary siRNA site for a distance of up to approximately 200–700 nt [17,18]. This mechanism can significantly increase the efficiency of RNAi. To investigate whether primary non-canonical sRNAs may also mediate formation of secondary transitive sRNAs, we examined sRNA read profiles within 200 nt zones on both sides of the dsRNA target. Our analysis did not reveal any tangible amounts of non-canonical reads (possibly apart from 20 nt sRNA which may be a product of PVY RNA degradation; see Figure 3A) in these zones suggesting a lack of transitivity (Figure 6B). Taken together, these data imply that non-canonical sRNAs induced by dsRNApvy do not trigger transitive amplification and spread of secondary siRNAs from the site of a primary sRNA target.

## 3. Discussion

External applications of dsRNA hold great potential as a new non-transgenic tool for crop protection and trait improvement because of its ability to selectively downregulate gene expression [21,22]. Such non-transgenic applications of dsRNA are becoming the method of choice due to their broad applicability, cost efficiency and low environmental impact [21,22,24]. However, foliar delivery of nucleic acids into plants requires them to overcome/cross barriers such as the cuticle, cell wall, low cell uptake and nuclease attack in order to achieve a robust and resilient whole-plant RNA-silencing phenotype [41,42]. 

In spite of recent advances, the molecular mechanisms underpinning the activities of externally applied dsRNAs in gene expression regulation, particularly from the perspective of controlling plant pests and pathogens, remain largely uncharacterised and challenging. It is widely postulated that these mechanisms are based on classical RNAi pathways involving DCLs, RdRPs, AGOs and other RNAi-specific components [24,26,27,28,29,30,31,32,33]. There is, however, no direct evidence to support this idea. sRNAs are principal hallmarks of RNA silencing. Therefore, as the first step in elucidating the mechanism of foliar-applied dsRNA-mediated action, we carried out comparative analysis of sRNAs generated in a classical virus-induced RNA-silencing system (PVY infection) in conjunction with foliar dsRNA application. Our results revealed three remarkable differences between the properties of sRNAs generated in these two systems. 

Firstly, unlike discrete canonical 21 nt and 22 nt long siRNAs cleaved from virus-derived dsRNA during natural virus infections, exogenous dsRNAs induce in potato the production of a non-canonical ladder-like set of sRNAs with a length of 18–30 nt (Figure 3). Interestingly, similar size distribution profiles have been described by Tabein et al. [30] and Rego-Machado et al. [31] when they applied exogenous dsRNA against tomato spotted wilt virus in *N. benthamiana* and tomato mosaic virus in tomato plants, respectively. Similar profiles were observed by Nityagovsky et al. [28], who exogenously applied dsRNAs designed against two different host endogenous genes in Arabidopsis plants. In all these cases, as well as in our own experiments, the formation of non-canonical sRNAs were accompanied by target-specific silencing. In contrast, Uslu et al. [36] did not detect any effect of GFP-specific dsRNA on GFP gene expression. The reasons for these discrepancies are not clear. Thus, our work confirms that externally applied dsRNA may induce in potato plants both formation of ladder like sRNAs species and sequence-specific virus suppression, suggesting that that these two processes are mechanistically interlinked. Given that DCLs are well-known to generate sRNAs of discrete sizes [35] these results may collectively account for a DCL-independent pathway for sRNA generation from external dsRNAs which can occur in at least in some plant species. Interestingly, DCL-independent routes for sRNA production probably also operate in some exogenous RNAi pathways. For example, such a route has been identified for biogenesis of sRNAs that play critical roles in gene regulation and transposon silencing through RdDM pathway in Arabidopsis. Normally in plants, RdDM requires 24 nt siRNAs which are processed from dsRNAs by DCL3 [43]. However, Ye et al. [44] described a distinct non-canonical class of functionally active sRNAs produced independently of DCLs in Arabidopsis which occurred as ladders of ~20 to 60 nt in length. These were derived from mainly transposons and intergenic sequences and transgenes via distributive 3′–5′ exonucleases [44]. Similar size distribution patterns of sRNAs involved in methylation was described by Yang et al. [45]. It is also possible that externally applied dsRNA (or its significant pool) is degraded non systematically either on the leaf surface or inside tissues. This may suggest that the dsRNA-based antiviral effect was mediated by a previously unrecognised mechanism which may be dependent on either dsRNA itself or non-canonical sRNAs, but is not based on RNAi. A second intriguing feature of non-canonical sRNAs identified in this work was their asymmetry in strand polarity with a strong bias towards antisense sequences. This asymmetry seems to be bizarre since DCLs are known to cleave dsRNA templates into siRNA duplexes, implying the equal presence of both sense and anti-sense siRNA strands. However, the preferential enrichment of one (either sense or anti-sense) of two siRNA strands in response to virus infection was also observed for many plant viruses [37,38,39], but this is usually explained by the effect of virus-specific factors, such as viral ssRNA itself or VSRs [37,38,39]. For example, bias towards sense strands fits well into the hypothesis that some pool of viral siRNAs may derive from regions of viral ssRNAs which exhibit substantial secondary (hairpin-like) structures [46]. Nevertheless, given that viral sense siRNAs do not always match local secondary structures in viral RNAs, some other mechanisms may be involved. VSRs, for instance, are able to modulate ratios between sense and anti-sense siRNAs. Indeed PVY HC-Pro VSR specifically down-regulates the accumulation of anti-sense secondary siRNAs while 2b VSR encoded by tomato aspermy virus supresses the accumulation of sense siRNA [39]. However, how two strands of externally applied dsRNApvy were distinguished from each other and selected (producing biased sRNA accumulation) in the absence of the virus remains puzzling. Although misinterpretation due to a potential bias in analysis of polarity of siRNA strands by means of computational algorithms cannot be completely excluded, it looks rather unlikely. Indeed, the preferential accumulation of one of two sRNA strands is not a common feature of any externally applied dsRNA. For example, analysis of sRNAs induced by potato virus S-specific dsRNA fragments carried out using the same computational algorithms did not reveal a similar level of asymmetry in strand polarity comparable with that shown in Appendix A. Thus, preferential accumulation of sRNAs with antisense polarity induced by dsRNApvy is probably determined by intrinsic properties of this particular dsRNA which have yet to be investigated. There are antisense guide sRNAs which are functionally active components in cleaving target viral RNAs, and selective loading of these into a specific AGO protein is favourably directed by the 5′ terminal nucleotide [10,11,40]. Most non-canonical antisense sRNAs induced by dsRNApvy are enriched in 5′A and 5′U, suggesting their association with AGO2 and AGO1 which are both involved in antivirus responses. However, it is unclear if and how these non-canonical size sRNAs can operate in degradation of virus RNA. 

Finally, it should be noted that effective silencing by RNAi usually depends on mechanisms that amplify siRNAs from target ssRNAs by RdRPs, which results in the production of massive amounts of transitive secondary siRNA that can spread systemically over the plant. Nevertheless, we were unable to detect any transitive secondary sRNAs corresponding to non-canonical size sRNAs induced by dsRNApvy even in the presence of the virus (when viral ssRNA was available). Consistent with this, we did not uncover any ability of the non-canonical sRNAs to move systemically on their own. Instead sRNAs are presumably generated in untreated leaves from dsRNAs which have some limited ability to move systemically. The abundance of such non-canonical sRNAs is low and they quickly disappear (in approximately a week).

Taken together, these remarkable dissimilarities in features between canonical siRNAs produced in natural systems (e.g., virus infections) and non-canonical sRNAs induced by foliar dsRNA applications may suggest the existence of an unsuspected earlier pathway or mechanism of sequence-specific dsRNA-mediated action which is different from RNAi. 

In plants, multiple natural RNAi pathways involve cascades of consecutive events that occur in a highly coordinated manner that typically include: (i) processing of long dsRNA or hpRNAs by DCLs into primary sRNA duplexes, (ii) amplification of sRNAs by RdRPs, (iii) methylation of sRNAs by HEN1 methyltransferase for their stabilisation, (iv) loading of one of the siRNA strands on an AGO protein possessing endonucleolytic activity, (v) target recognition through siRNA base pairing and (vi) cleavage of the target by the AGO’s endonucleolytic activity. These steps are highly compartmentalized and provide machinery for sorting sRNAs into distinct RNAi pathways to predetermine their biological function. Intuitively, we could hypothesize that sites in which dsRNA/hpRNAs (e.g., virus replication sites) are produced, may mechanistically link to RNAi-related compartments, and therefore endogenously produced dsRNA/hpRNA can automatically enter a certain RNAi pathway. This pathway then may operate like a conveyor system in which products of the preceding step are moved for consuming in a subsequent step, for example, passing primary sRNAs generated during DCL-based dsRNA cleavage to the sRNA amplification step, and so on. 

In contrast to most endogenously produced dsRNAs, external dsRNAs in our experiments were cleaved into non-canonical sRNAs by an unknown (yet to be elucidated) DCL-independent mechanism. It is unclear if or how AGO proteins could load non-canonical sRNAs of sizes (18–30 nt) not normally associated with AGO to form functionally active RISCs. One possibility is that a small fraction of external dsRNA was processed by DCL2 and DCL4 into AGO2 and AGO1-compatible siRNA duplexes with lengths of 22 nt and 21 nt, respectively, and that this particular fraction may be responsible for virus suppression as was suggested by Nityagovsky et al. [28] for RNA silencing of endogenous genes. This suggestion is consistent with data presented by Necira et al. [26] indicating that antivirus activity of topically applied dsRNA in *N. benthamiana* requires DCL2 and DCL4. At the same time, it should be noted that the vast majority of sRNAs generated by external dsRNA have non-canonical AGO-incompatible sizes. Other details related to activities of external dsRNAs are also rather obscure. Despite these limitations and gaps in knowledge, serious advances have been made in foliar applications of dsRNA [21,22,24] for crop improvement and protection, but for further technological developments more research is required to better understand the precise mechanisms induced by external dsRNAs, and some future directions and research priorities are listed in below in the Concluding remarks section.

## 4. Materials and Methods

### 4.1. Virus, Plants and Growth Conditions

PVY (NTN strain, PVY-NTN) was propagated in potato plants (*Solanum tuberosum* L.; cv. Manhattan). Potato plants (*Solanum tuberosum* L.; cv. Indigo) were grown in soil from in vitro-propagated plantlets. Two-week-old Indigo plants were used for experiments. Three fully expanded leaves per plant and four plants per each experimental condition were treated with dsRNA solution or buffer, and were inoculated the next day with buffer or PVY (crude extract from infected plants) and left to grow in a controlled environment chamber (Pol-Eko-Aparatura, Wodzisław Śląski, Poland) which was set at a photoperiod of 16/8 h day/night with a relative humidity of 40% and a light fluence of 250 µmol m^−2^ s^−1^.

### 4.2. Production and Purification of dsRNA (dsRNApvy)

cDNA corresponding to the conservative fragment of the PVY replicase gene [47], which has a length of 500 bp (nts 7739–8238; Gene Bank accession number OR545670), and cDNA corresponding to the fragment of the PVS RdRP and TGBp1 viral gene region, which has a length of 499 bp (nts 5874–6373; Gene Bank accession number LN851189) were synthesized by Evrogen (Moscow, Russia) and cloned into the plasmid vector L4440 (plasmid 1654; Addgene Watertown, MA, USA). This vector contains two T7 promoters in an inverted orientation that flanks the multiple cloning sites. The recombinant L4440 vector was transformed into *Escherichia coli* strain HT115 (DE3) via standard transformation procedures [48]. This strain does not produce RNase III, a dsRNA degrading enzyme. T7 RNA polymerase-mediated transcription was induced with isopropyl β-d-1-thiogalactopyranoside (IPTG; 1 mmol/L). The bacterial cultures were centrifuged at 5000× *g* for 10 min at 4 °C. After discarding the supernatant, the bacterial pellet was resuspended in extraction buffer (5% sucrose, 50 mM EDTA, 50 mM Tris-HCl pH 8.0, 0.75 M NH_4_Cl, 0.5% Triton X-100) containing lysozyme in a final concentration of 100 µg/mL (for total RNA extraction), heated to 65 °C for 5 min and centrifuged at 20,000× *g* for 10 min. The supernatant was mixed with 0.7 volume of isopropanol, stirred and centrifuged at 10,000× *g* for 10 min. Total RNA pellet was washed twice with 70% ethanol, dried at room temperature for 30 min and then, to increase the yield of correctly paired RNA duplexes, was dissolved in 10 mM Tris-HCl pH 7.5, 2.5 mM MgCl_2_, 0.1 mM CaCl_2_, heated for 5–10 min at 95 °C and gradually cooled down to room temperature (over a period of no less than 90 min). The purified dsRNA was quantified using a NanoDrop 2000 spectrophotometer and examined in 1.2% agarose gels stained with BrightGreenDNA (Sileks, Moscow, Russia) under UV light.

### 4.3. Exogenous dsRNA Application for Plant Protection against Virus Infection

The dsRNApvy solution in Milli Q water (40 µg per leaf) was applied to plant leaves in the presence of 1000-fold diluted surfactant (Silwet^®^ L-77; Momentive, Niskayuna, NY, USA) using a mechanical pipette. Drops of dsRNA solution were evenly distributed over the leaf surface by spreading with a gloved finger. Buffer (including surfactant)-treated healthy plants were used as control. Then, 24 h after dsRNA or buffer application, the same leaves were challenged by mechanical inoculation by the virus.

### 4.4. Plant RNA Extraction and Real Time Quantitative RT-PCR (RT-qPCR)

Leaf tissues (1 to 2 g) were frozen in liquid nitrogen and ground to a fine powder in a mortar and pestle, and total RNA was extracted using TRI REAGENT according to the manufacturer’s recommendations (Sigma-Aldrich, St. Louis, MO, USA). RNA was suspended in 50 µL DEPC-treated water. DNase-treated RNA was reverse-transcribed into cDNA using the SuperScript^TM^ First-Strand Synthesis System for RT-PCR (Invitrogen, Carlsbad, CA, USA) and analysed by SYBR green-based real-time PCR as described earlier [33]. The Ct values for PVY RNA were normalized using two internal reference genes encoding *StEF-1α* [49] and cytochrome c oxidase subunit 1 (*StCOX*) [50]. 

### 4.5. RNA Sequencing and Data Analysis

Preparation of RNA and sRNA libraries and next-generation sequencing were performed by CeGaT (Tuebingen, Germany). Briefly, RNA sequencing libraries were prepared using TruSeq Stranded Total RNA in conjunction with a Ribo-Zero kit (Illumina, San Diego, CA, USA). The sequencing of the library pool was carried out using the Illumina NovaSeq 6000 platform with a paired-end sequencing strategy of 1 × 100 bp. Small RNA was prepared for sequencing using the NEXTFlex Small RNA-Seq v3 kit (Bio Scientific, Gymea, Australia) which were then sequenced using the Illumina NovaSeq 6000 with a single-read length of 50 nucleotides. The libraries were de-multiplexed, followed by adapter trimming. The quality of the FASTQ files was analyzed with FastQC (version 0.11.5-cegat) [51]. After low-quality filtering and removal of reads shorter than 18 and longer than 30 nucleotides with prinseq-lite [52], clean reads were mapped to the PVY-NTN genome (OR545670) using bowtie2 version 2.3.5.1 [53], with zero mismatches. Mapping results were visualized using MISIS-2 [54], Microsoft Excel and Python version 3.11.3 custom scripts. Samtools package version 1.10 was used for operations with sam/bam files [55]. To analyze viral siRNAs, the reference sequences of PVY-NTN genomes were used to map 18–30 nt reads from each library. The sorted sRNAs were then counted by size, polarity, and 5′-terminal nucleotide identity (5′A, 5′C, 5′G and 5′U) using in-house scripts. Sequence data have been submitted to the Sequence Read Archive (SRA) with the BioProject accession PRJNA1018135.

## 5. Conclusions

Global sustainability policies emphasize the need to replace contentious pesticides with safe, efficient and cost-effective alternatives to ensure sustainable food production. dsRNA-mediated biocontrol can be applied using two main approaches: *in planta* delivery through transgenic (GMO) crops or exogenous application of formulated RNA-based products. Foliar non-transgenic applications of dsRNA for crop protection against viruses, fungal pathogens and insect pests, and for regulation of endogenous gene expression have come to prominence due to higher selectivity and better safety profiles (less mobile through the soil, less persistent and less toxic) compared with controversial chemical pesticides [23,41,56,57]. This approach is also favoured because plants treated with dsRNAs are not considered as GMOs, and therefore don’t carry the public perception risks often associated with GMOs such as negative safety, environmental and ecological impacts. These RNA-based products can also be directly applied using current agricultural practices, such as spray applications, trunk injection for trees and seed soaking. The commercial interest in RNA applications has also risen due to the development of biotechnology tools which can mass produce dsRNA at low cost for the agribusiness [57,58].

However, along with promising results in recent studies, various factors limiting the efficiency of dsRNA-based applications have been recognized in plants. To penetrate plant cells, foliar-applied dsRNAs must avoid nuclease degradation, traverse the plant cuticle and cell wall and plasma membrane [42]. Some progress has been achieved in the development of appropriate formulation technologies for the delivery of dsRNA molecules into plants to target crop pests or pathogens [29,30]. Formulations and nanoplatforms for delivery are generally designed to improve dsRNA stability and ensure effective penetration of dsRNA into plant cells. Among the nanoplatforms that could be potentially used for dsRNA foliar delivery via spraying are silica nanoparticles, layered double hydroxides (nanoclay), carbon-based materials (carbon dots and single-walled nanotubes), chitosan and cell-penetrating peptides [29,30,59].

It should also be noted that even if an external dsRNA enters a plant cell, many other barriers seem to exist that prevent association of dsRNA with the correct RNAi pathway [28,30,42]. Data presented in this and other [28,30,31,36] papers show that external application of dsRNA in plants can lead to generation of non-canonical 18–30 nt sRNAs suggestive of a DCL-independent process. Thus, it would be conceivable to elucidate mechanisms of their biogenesis and antiviral action, for example, using *dcl* and *ago* mutants. It would also be useful to use TrAP-R technology to study if the “non-canonical” sRNAs could successfully associate with AGO proteins. 

It would also be useful to compare the effectiveness of the non-canonical sRNAs versus classical 21 nt and 22 nt sRNAs in terms of antiviral activity and crop protection. Although direct comparison is not possible, the activity of the same dsRNA fragment could be examined in HIGS and virus-induced gene silencing (VIGS) (both produce 21 nt and 22 nt sRNA species) versus external dsRNA applications (non-canonical sRNAs). However, even before such analysis, it is clear that HIGS and VIGS technologies have some significant limitations such as their time-consuming nature and also the likely public concerns regarding genetically modified transgenic plants. With regard to external dsRNA applications, limitations may include a short duration of action, which may be overcome by repetitive treatments or by using stabilizing formulations. External dsRNA applications have advantages over classical RNAi technologies as it does not require permanent irreversible genetic modification and can operate in real time at specific timepoints to deliver desirable outcomes without impacting other plant functions such as growth and development; decreasing the risk of any unintended effects.

Efficient transitive amplification and mobility of siRNA is a prerequisite of robust RNAi defence responses. Therefore, future studies should examine why these processes do not take place in the case of dsRNA applications and how they may be activated.

Detailed analysis of dsRNA targets with a focus on their functional relevance is also needed to ascertain their impact on viral replication, spread and overall crop protection. To that end, we have recently initiated research in order to analyse real natural plant virus populations in different geographical and climatic zones of Russia [60], helping to identify potential common targets in diverse virus populations.

Thus, development of new approaches to increase the incorporation of foliarly applied dsRNAs into highly orchestrated plant host pathways would be crucial for the feasibility of systemic crop protection.

## Figures and Tables

**Figure 1 ijms-24-15769-f001:**
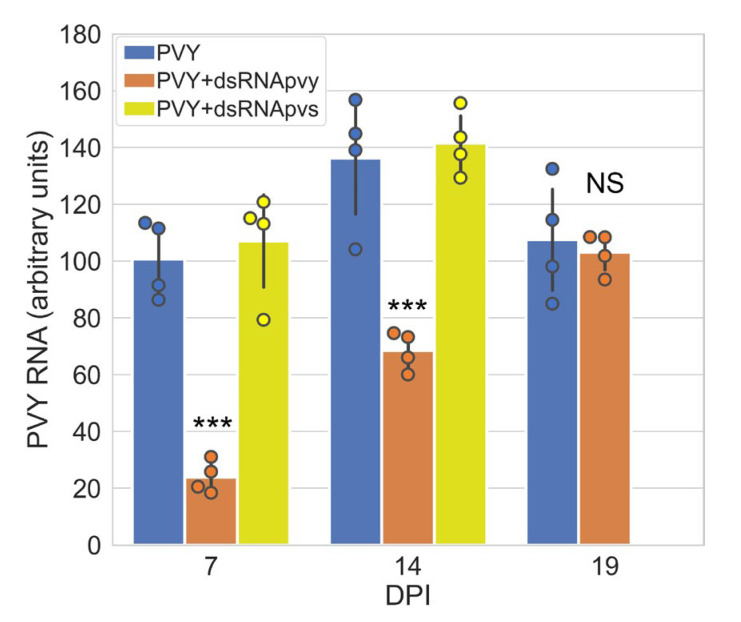
Accumulation of PVY RNA (measured using RT-qPCR) in systemically infected leaves of potato plants cv. Indigo pre-treated with dsRNApvy, dsRNApvs and buffer (mock, PVY) over 7–19 days post infection (dpi) time periods as shown. PVY RNA expression levels were normalized to those of internal controls, *StEF-1α* and *StCox*. Statistical analysis was performed on four independent biological replicates. Data are mean ± SD. ANOVA and Tukey’s HSD post hoc tests were performed on the RT-qPCR data. *** *p* < 0.001; NS, not significant.

**Figure 2 ijms-24-15769-f002:**
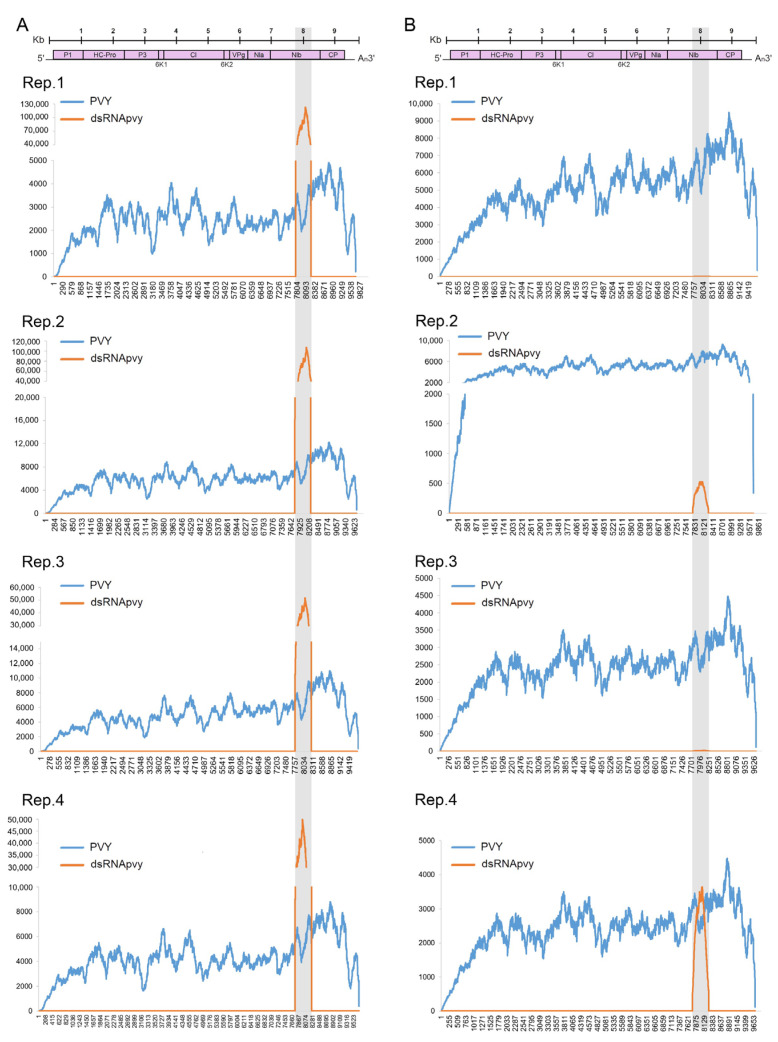
Distribution of dsRNApvy in plants exogenously treated with dsRNA RNA. Sequencing read coverage across the PVY-NTN genome in samples from ds-RNApvy treated (**A**) and untreated (**B**) leaves determined by HTS. Coverage of PVY sequences in inoculated (**A**) and uninoculated systemically infected (**B**) leaves of PVY-infected plants is shown as control. Different graphs represent individual replicates (Rep) as indicated. The organization of the PVY genome is shown schematically above the graphs. The PVY target region is highlighted in grey.

**Figure 3 ijms-24-15769-f003:**
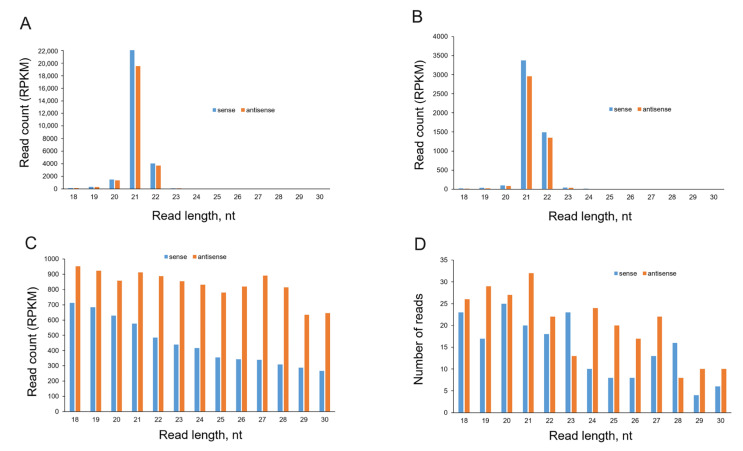
Size and polarity profiles of PVY-specific sRNAs. Reads obtained from inoculated (**A**) and uninoculated systemically infected (**B**) leaves of PVY-infected plants and from treated (**C**) leaves of dsRNApvy-treated plants. These were mapped with zero mismatches to the reference sequence (PVY-NTN). The mapped reads were sorted by size (from 18 nt to 30 nt) and polarity (sense, anti-sense) and counted in Reads Per Kilobase Million (RPKM) in each library. The data represent averages of four replicates per each treatment. (**D**) Reads that were obtained from untreated leaves of dsRNApvy-treated plants that also contained dsRNA. The total numbers of mapped reads are indicated. The resulting counts are plotted as bar graphs and color-coded as shown.

**Figure 4 ijms-24-15769-f004:**
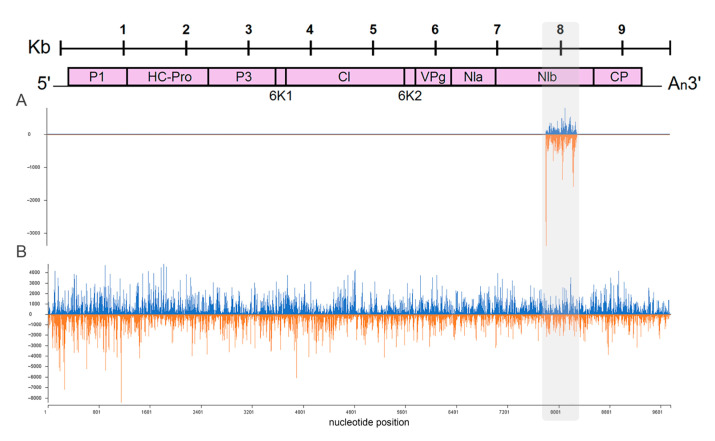
Representative images of single-nucleotide resolution maps of dsRNApvy- (**A**) or PVY-derived (**B**) sRNAs. sRNAs were mapped to the PVY-NTN reference genome. Bars above the axis represent sense reads starting at each respective position; those below represent antisense reads ending at the respective position. The organization of the PVY genome is shown schematically above the histograms. The PVY target region is highlighted in grey.

**Figure 5 ijms-24-15769-f005:**
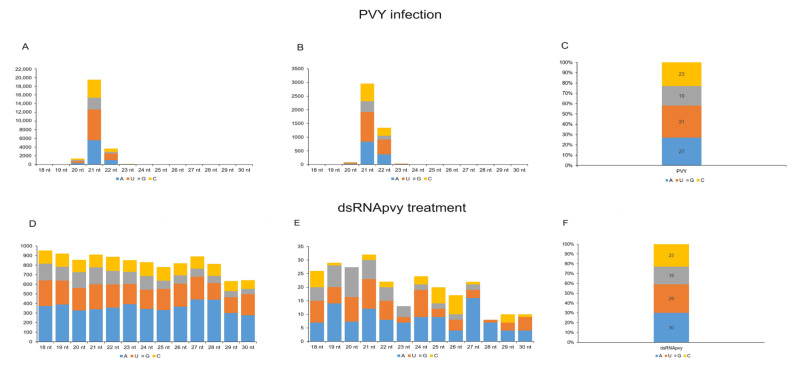
The 5′-nucleotide profiles of major size classes of anti-sense dsRNApvy- and PVY-derived sRNAs. The sRNA reads from inoculated (**A**) or systemically infected (**B**) leaves of PVY-infected plants or treated (**D**) and untreated (**E**) leaves of plants which were administered with dsRNApvy were mapped with zero mismatches to reference PVY-NTN sequences. The mapped sRNAs were then sorted by size and 5′-terminal nucleotide identity (5′A, 5′C, 5′G and 5′U). The frequencies of each 5′-nucleotide (in % of total) for each major size class are plotted as bar graphs and colour-coded. (**C**,**F**), A,U,G, C content distribution in the PVY genome and in the dsRNApvy fragment, respectively.

**Figure 6 ijms-24-15769-f006:**
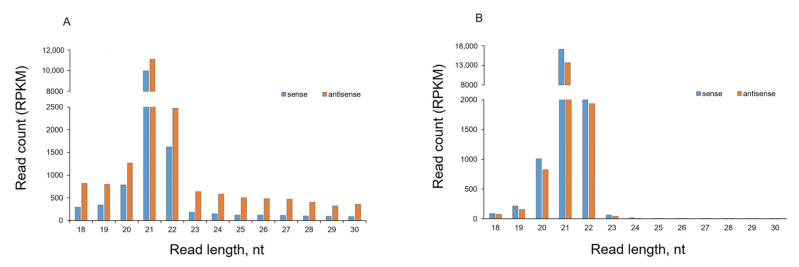
Size and polarity profiles of PVY-specific sRNAs detected in dsRNApvy-treated leaves of PVY-infected plants. (**A**) sRNAs mapped to the dsRNApvy target zone of PVY genome (nts 7739–8238). (**B**) sRNAs mapped with zero mismatches to the 200 nt zones upstream (nts 7539–7738) and downstream (nts 7939–8138). The mapped reads were sorted by size (from 18 nt to 30 nt) and polarity (sense, anti-sense) and counted in Reads Per Kilobase Million (RPKM) in each library. The resulting counts are plotted as bar graphs and color-coded as indicated. The data represent averages of four replicates.

## Data Availability

The data supporting the findings of this study are available within the article and in the NCBI database, BioProject accession PRJNA1018135.

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
