# Peer review of "A Non-Canonical Pathway Induced by Externally Applied Virus-Specific dsRNA in Potato Plants"

_ijms, 2023, doi:10.3390/ijms242115769_

Round 1
Reviewer 1 Report
Comments and Suggestions for Authors
Samarskaya et al addressed the molecular mechanism of RNAi mediated virus defense upon exogenous dsRNA applications. The article is certainly of great interest to RNAi researchers as well as researchers developing sustainable pesticide alternatives to classical chemical pesticides. Employing sRNA-sequencing as a hallmark of RNAi in exoRNA applications is a powerful approach. However, the findings are not well supported and the interpretation of the data is misleading.
First of all, the authors show that PVY accumulation is mitigated by the use of PVY-specific dsRNA. However, it is clear that it may not be a specific effect. Rather the pH, dsRNA stimulation or additional factors may underlie this effect. Therefore without a non-target dsRNA use, it is impossible to conclude that dsRNApvy can hinder the virus accumulation.
In the second figure, the authors performed the sRNA seq. experiment. In the second column, sRNA composition of the systemic (non-sprayed) leaves were depicted. In only 2/4 samples, they detect the sRNAs mapping to the dsRNApvy. The standard deviation of the dsRNApvy reads indicates that it is very likely to be a contamination (due to while spreading the dsRNA with a glove) rather than a physiological movement of the dsRNApvy to the systemic parts. The claim of mobile exoRNAs require further contamination controls. For example, non-sprayed leaves are collected, right after the spraying experiment (which excludes systemic movement) or RNase treatment of the contaminating RNA on the leaf surface. In addition, it is also clear that there is no reverse correlation between the read counts of dsRNApvy and PVY reads. If dsRNA really blocks the PVY -as stated in Figure 1- then the 40k dsRNA reads in Rep3 and Rep4, would have given the highest PVY reads, which is not the case (the highest PVY reads are in Rep2).
In Figure 3, the authors present the non-canonical sRNAs 18-30nt as a new kind of processed RNAs. However, they were previously shown in Uslu et al 2020 and Nityagoxsky et al 2022. It can simply be the degradation product of the dsRNA, which remains on the leaves, rather than any “non-canonically” processed RNAs.
In Figure 5C, the authors claim that there is the 5’ nucleotide bias -regardless of the size of the sRNA. However, they have not normalized it to the base-pair distribution of dsRNApvy. I could not find the Gene Bank accession number OR545670) but I used another accession numbers to check the PVY replicase composition: It is 30-35% A, 28-32%U -very similar to the 5’-nucleotide distribution given by in Figure 5C. Therefore, it is also an indirect proof of simple degradation, rather than any processing of the exo dsRNA.
One suggestion is that the authors use TrAP-R instead of simple sRNA sequencing to show that the “non-canonical” sRNAs are associated with AGO proteins.
To sum up, the authors used sRNA-seq to assess the mechanism of exo-RNA mediated virus resistance. However, the findings are not new (18-30nt non-canonical RNAs) and they have not added any special analysis to show that these RNAs are not simple RNA degradation products on the leaf surface /or systemic leaf contaminations but they are physiological. It is impossible to reach their conclusions, simply by the given data. Rather, the 5’nucleotide analysis supports that the sRNAs were simply degradation products limited to the leaf surface.
Comments on the Quality of English LanguageThe paper is clearly understandable. There is only a moderate amount of mistakes, which can be corrected at the editing phase.
Author Response
Reviewer 1.
Samarskaya et al addressed the molecular mechanism of RNAi mediated virus defense upon exogenous dsRNA applications. The article is certainly of great interest to RNAi researchers as well as researchers developing sustainable pesticide alternatives to classical chemical pesticides. Employing sRNA-sequencing as a hallmark of RNAi in exoRNA applications is a powerful approach. However, the findings are not well supported and the interpretation of the data is misleading.
We would like to thank the reviewer for their generally positive comment and will try to clarify what is misleading (see below).
First of all, the authors show that PVY accumulation is mitigated by the use of PVY-specific dsRNA. However, it is clear that it may not be a specific effect. Rather the pH, dsRNA stimulation or additional factors may underlie this effect. Therefore without a non-target dsRNA use, it is impossible to conclude that dsRNApvy can hinder the virus accumulation.
It seems to us that this is the main comment and our response to it may clarify many of the remaining uncertainties raised. It is common practise amongst researchers to include different non-target dsRNAs alongside target dsRNAs as controls. We included such controls in our experiments and found that only the dsRNApvy had highly nucleotide sequence specific activity. We did not include these data in the original text, given that various other reports demonstrated specific antiviral effects of the externally applied dsRNA. We have now included these control data, whereby dsRNA designed to the sequence of potato virus S genome was used to support our suggestion about specific action of dsRNApvy (Figure 1, lines 163-164).
In the second figure, the authors performed the sRNA seq. experiment. In the second column, sRNA composition of the systemic (non-sprayed) leaves were depicted. In only 2/4 samples, they detect the sRNAs mapping to the dsRNApvy. The standard deviation of the dsRNApvy reads indicates that it is very likely to be a contamination (due to while spreading the dsRNA with a glove) rather than a physiological movement of the dsRNApvy to the systemic parts. The claim of mobile exoRNAs require further contamination controls. For example, non-sprayed leaves are collected, right after the spraying experiment (which excludes systemic movement) or RNase treatment of the contaminating RNA on the leaf surface. In addition, it is also clear that there is no reverse correlation between the read counts of dsRNApvy and PVY reads. If dsRNA really blocks the PVY -as stated in Figure 1- then the 40k dsRNA reads in Rep3 and Rep4, would have given the highest PVY reads, which is not the case (the highest PVY reads are in Rep2).
Indeed, in only 2/4 samples we detected dsRNApvy in non-treated leaves. We cannot exclude completely that this is due to contamination, however this seems to us much less likely than physiological movement in planta, because the “non-treated” leaves we used for analysis did not exist at the time of treatment but rather emerged at a later stage (there were no leaves to collect and analyse right after treatment). We have now included these clarifications in the text, and also added a statement mentioning that we cannot completely exclude contamination (lines 217-218).
With regards to the second part of the comment, diagrams in Figure 2 represent read coverage maps for treatments with dsRNApvy only (without PVY) just to confirm that they exclusively matched the PVY genome region, which was used for dsRNA design and construction. PVY (PVY only without dsRNA) maps were obtained from separate samples only as controls and should not be compared with dsRNA maps in terms of PVY suppression (lines 197 -202).
In Figure 3, the authors present the non-canonical sRNAs 18-30nt as a new kind of processed RNAs. However, they were previously shown in Uslu et al 2020 and Nityagoxsky et al 2022. It can simply be the degradation product of the dsRNA, which remains on the leaves, rather than any “non-canonically” processed RNAs.
The main point is that the modified Figure 1 clearly shows that an anti-PVY effect is triggered specifically via externally applied dsRNApvy (but not with non-specific dsRNA), which in accordance with previous reports, suggesting that dsRNAs induces PVY suppression through a mechanism similar to RNAi. Moreover, we clearly show that dsRNA is processed into 18-30 nt sRNAs by a yet unknown mechanism, and therefore we are functionally and mechanistically connecting these two events. When we described 18-30 nt sRNAs in the original version, we referred to other previous works: Tabein et al. [24] and Nityagovsky et al. [22]. Unfortunately, Uslu et al. (2020) was missed. In the case of Nityagovsky et al. [22] and Uslu et al. (2020), the authors also showed formation of 18-30 nt sRNA ladders from dsRNA. However, the results were contradictory: while Nityagovsky et al. [22] showed silencing effects on two endogenic/transgenic plant genes including GFP, Uslu et al. (2020) did not detect any effect on GFP expression. Tabein et al. [24] and Rego-Machado et al. [25] also showed that formation of ladder-like sRNA size distribution in N. benthamiana and tomato plants correlated with virus suppression. Reasons for these discrepancies are not clear. Our work also confirms that externally applied dsRNA induces in potato plants the formation of ladder-like sRNAs species and also a sequence-specific virus suppression, suggesting that that these two processes are mechanistically interlinked. The text has been modified accordingly (lines 246 – 257).
In Figure 5C, the authors claim that there is the 5’ nucleotide bias -regardless of the size of the sRNA. However, they have not normalized it to the base-pair distribution of dsRNApvy. I could not find the Gene Bank accession number OR545670) but I used another accession numbers to check the PVY replicase composition: It is 30-35% A, 28-32%U -very similar to the 5’-nucleotide distribution given by in Figure 5C. Therefore, it is also an indirect proof of simple degradation, rather than any processing of the exo dsRNA.
I believe that this comment may have arisen due to a misunderstanding caused by incorrectly used phrases in our text. What we intended to state is: Non-canonical antisense (18 nt – 30 nt) sRNA species detected in dsRNApvy treated plants showed significant variations in patterns of 5’ nucleotide distribution depending on the length of the sRNA species. The distribution pattern of nucleotides at the 5’-ends of sRNAs does not generally correlate with the ratio between nucleotides in the antisense fragment of dsRNApvy. For example, although in some of the cases (e.g. 21 nt and 22 nt fragments) which were mentioned by the reviewer, distribution between A and U actually resemble A:U ratio in the dsRNApvy fragment, while C/G proportions at the 5’ ends of the sRNAs (~ 1:3) is quite different from that of the whole dsRNA fragment (~ 1:1). With some other sRNA species, patterns of 5’-terminal nucleotide distribution are even much less correlated with the A/U/C/G ratio in the dsRNA fragment. For instance, with 27 nt and 28 nt species, “A” significantly prevails over all other nucleotides (reaching ~ 50% in treated leaves); in untreated leaves, 20 nt and 23 nt species do not contain C at the 5’-end at all; 28 nt species contains neither C, nor G at the 5’ end. These data so far do not propose any sequence-selective recognition of nucleotides for processing/decay of externally applied dsRNAs. The mechanism(s) underpinning this process requires further investigation. Interestingly, any specificity (bias) of DCL cleavage of dsRNAs is also unknown. However, despite the variable distribution of nucleotides at the 5’-ends of the non-canonical sRNA, they still have potential affinity (sufficient number of A, U or C) to the same repertoire of AGO proteins (AGO1, AGO2 and AGO5) as PVY-induced siRNAs. At the same time, it is unclear if most sRNAs found in dsRNApvy treated plants, which are actually non-canonical sRNAs (with atypical size) are able to form RISCs with AGO proteins”. The text and Figure 5 have been modified accordingly and Supplementary Figure S2 has been added (lines 314 -333).
One suggestion is that the authors use TrAP-R instead of simple sRNA sequencing to show that the “non-canonical” sRNAs are associated with AGO proteins.
We would like to thank the reviewer for providing us with this interesting idea. We will definitely follow their advice in the future along with other approaches, such as the use of dcl and ago mutants; which will be invaluable to expand upon our good experimental system in potato and prepare a toolbox to assess the mechanistic aspects of the specific and broad spectrum DCLs, AGOs and RdRPs in potato crops.
To sum up, the authors used sRNA-seq to assess the mechanism of exo-RNA mediated virus resistance. However, the findings are not new (18-30nt non-canonical RNAs) and they have not added any special analysis to show that these RNAs are not simple RNA degradation products on the leaf surface /or systemic leaf contaminations but they are physiological. It is impossible to reach their conclusions, simply by the given data. Rather, the 5’nucleotide analysis supports that the sRNAs were simply degradation products limited to the leaf surface.
In conclusion, we would like to thank the reviewer once again for valuable suggestions which as I hope significantly helped us to improve the paper. We have now addressed all of the comments, and our responses are listed above. We have also added many new citations, including those mentioned by the reviewer.
Reviewer 2 Report
Comments and Suggestions for Authors#The study presents intriguing findings regarding the non-canonical sRNA biogenesis pathway induced by exogenous application of PVY dsRNA, but more comprehensive mechanistic insights are needed to understand this unexpected pathway and its implications fully.
#The study assumes similarities between dsRNA-mediated antivirus RNA silencing and natural RNAi-based defense against RNA-containing viruses without providing direct evidence. Further investigations should validate this assumption and elucidate the precise mechanistic connections.
#While the safety and environmental benefits of non-transgenic RNA silencing approaches are highlighted, a more in-depth discussion of potential limitations, challenges, and ethical considerations associated with using exogenous dsRNA for crop protection is warranted.
#The study focuses on a specific virus (PVY), and it would be beneficial to broaden the scope to investigate how the non-canonical sRNA generation pathway may vary with different types of viruses, potentially shedding light on virus-specific mechanisms.
#The lack of systemic movement and transitive amplification of the non-canonical sRNAs is intriguing, but further research is needed to explore the underlying molecular and cellular processes that govern their behavior and functionality.
#The implications of the non-canonical sRNA biogenesis pathway on crop protection are mentioned, but a deeper discussion on potential practical applications, advantages, and limitations in the context of crop protection strategies is necessary.
#The study could benefit from a comparison of the effectiveness of the non-canonical sRNAs versus traditional 21 and 22 nt sRNAs in terms of antiviral activity and crop protection, providing a more thorough understanding of their potential utility.
#The sequencing analysis provides valuable data on sRNA profiles, but a more detailed analysis of their targets and functional relevance is needed to ascertain their impact on viral replication, spread, and overall crop protection.
#The paper should discuss the stability and persistence of the non-canonical sRNAs in the plant system and whether their presence over extended periods could pose any unintended effects or complications.
#Future directions and research priorities in the field of RNAi-based crop protection should be outlined, emphasizing what gaps in knowledge this study helps to fill and what questions still need to be addressed to advance this promising approach.
#The article lacks proper citations and references to support the claims and information presented. A more rigorous referencing approach is necessary to ensure the accuracy and credibility of the content.
Suggested references: 1. Sundaresha, S. et al., 2022, Spraying of dsRNA molecules derived from Phytophthora infestans, along with nano clay carriers as a proof of concept for developing novel protection strategy for Potato late blight. Pest Management Science, 78(7): 3183-3192. https://doi.org/10.1002/ps.6949
2. Sundaresha S. et al., 2022. In Vitro Method for Synthesis of Large-Scale dsRNA Molecule as a Novel Plant Protection Strategy. In: Mysore, K.S., Senthil-Kumar, M. (eds) Plant Gene Silencing. Methods in Molecular Biology, vol 2408. Humana, New York, NY. https://doi.org/10.1007/978-1-0716-1875-2_14
#The language used is generally clear, but the lack of specific examples and detailed explanations makes it challenging for readers who are not experts in the field to fully grasp the concepts presented.
Comments on the Quality of English LanguageMinor editing of English language required.
Author Response
#The study presents intriguing findings regarding the non-canonical sRNA biogenesis pathway induced by exogenous application of PVY dsRNA, but more comprehensive mechanistic insights are needed to understand this unexpected pathway and its implications fully.
I would like to thank the reviewer for their generally positive comment. We do agree with them that more comprehensive mechanistic insights are needed and that will be the challenge for the future. In particular, we are planning to use dcl and ago mutants to elucidate the mechanism(s) underpinning the activity of non-canonical sRNAs. In addition, we intend to use the TrAP-R technology to show whether the “non-canonical” sRNAs are associated with AGO proteins. However, I should note that for this purpose, we need to prepare an appropriate toolbox for potato crops (Solanum tuberosum contains a specific and broad spectrum of DCLs, AGOs and RdRPs). The current work is a very important step forward as it has allowed us to establish a good experimental system in potato.
#The study assumes similarities between dsRNA-mediated antivirus RNA silencing and natural RNAi-based defense against RNA-containing viruses without providing direct evidence. Further investigations should validate this assumption and elucidate the precise mechanistic connections.
Indeed, similarities between dsRNA-mediated antivirus RNA silencing and natural RNAi-based defense against RNA-containing viruses were assumed by numerous reports in the literature based on the sequence-specific mechanism of the antiviral effect of externally applied dsRNA. We have now added data on the null effect of non-specific dsRNA (complementary to a fragment of potato virus S genome) on PVY accumulation (see Figure 1, and lines 163 -164). This also suggests that dsRNApvy specifically protects potato plants from PVY. However, our paper also presents differences with classical RNAi such as DCL-independent generation of non-canonical sRNAs; inability of sRNAs to move systemically and induce transitive synthesis of secondary RNAs. We do agree that further investigations, some of which were described above, should be performed to elucidate the precise mechanisms underpinning the phenomenon.
#While the safety and environmental benefits of non-transgenic RNA silencing approaches are highlighted, a more in-depth discussion of potential limitations, challenges, and ethical considerations associated with using exogenous dsRNA for crop protection is warranted.
These aspects have now been discussed briefly, taking into account that this is a Research article rather than the Review paper (lines 576 - 591).
#The study focuses on a specific virus (PVY), and it would be beneficial to broaden the scope to investigate how the non-canonical sRNA generation pathway may vary with different types of viruses, potentially shedding light on virus-specific mechanisms.
There are at least two other research works published, including Tabein et al. [24] and Rego-Machado et al. [25], which suggest that non-canonical sRNAs (or part of them) may trigger antiviral defence. Moreover, externally applied RNAs may suppress expression of endogenous genes as has been described by Nityagovsky et al. [22]. These have now been explained in more detail and integrated into the manuscript to illustrate the generalized effect of dsRNAs (lines 400 -411).
#The lack of systemic movement and transitive amplification of the non-canonical sRNAs is intriguing, but further research is needed to explore the underlying molecular and cellular processes that govern their behavior and functionality.
We do agree, and this point has now been emphasized in the text (lines 624 -626).
#The implications of the non-canonical sRNA biogenesis pathway on crop protection are mentioned, but a deeper discussion on potential practical applications, advantages, and limitations in the context of crop protection strategies is necessary.
We do agree and additional notes have now been included in the text (lines 576 -591).
#The study could benefit from a comparison of the effectiveness of the non-canonical sRNAs versus traditional 21 and 22 nt sRNAs in terms of antiviral activity and crop protection, providing a more thorough understanding of their potential utility.
We do agree and additional comments have now integrated into the text (lines 611-621).
#The sequencing analysis provides valuable data on sRNA profiles, but a more detailed analysis of their targets and functional relevance is needed to ascertain their impact on viral replication, spread, and overall crop protection.
We absolutely agree. In order to develop more pragmatic approaches for rational design of dsRNA targets in virus genomes, we have recently initiated a new research direction to analyse real viral populations in different regions (Int. J. Mol. Sci. 2023, 24(19), 14833; https://doi.org/10.3390/ijms241914833). Additional notes have now been included into the text (lines 625 -629).
#The paper should discuss the stability and persistence of the non-canonical sRNAs in the plant system and whether their presence over extended periods could pose any unintended effects or complications.
We have done this (lines 616 - 624).
#Future directions and research priorities in the field of RNAi-based crop protection should be outlined, emphasizing what gaps in knowledge this study helps to fill and what questions still need to be addressed to advance this promising approach.
We have done this (see new version of the Concluding remarks section).
#The article lacks proper citations and references to support the claims and information presented. A more rigorous referencing approach is necessary to ensure the accuracy and credibility of the content.
Many citations have been added to the text including those recommended by the reviewer.
Round 2
Reviewer 1 Report
Comments and Suggestions for Authors
Most of the issues that I highlighted have been addressed and the strong claims of the paper have been rightfully toned down. The report and the data presented by the authors are valuable and it is worth publishing them. However, the following concepts should be rephrased to avoid any misunderstanding. These points are indicated below in green (and italic).
Samarskaya et al addressed the molecular mechanism of RNAi mediated virus defense upon exogenous dsRNA applications. The article is certainly of great interest to RNAi researchers as well as researchers developing sustainable pesticide alternatives to classical chemical pesticides. Employing sRNA-sequencing as a hallmark of RNAi in exoRNA applications is a powerful approach. However, the findings are not well supported and the interpretation of the data is misleading.
We would like to thank the reviewer for their generally positive comment and will try to clarify what is misleading (see below).
First of all, the authors show that PVY accumulation is mitigated by the use of PVY-specific dsRNA. However, it is clear that it may not be a specific effect. Rather the pH, dsRNA stimulation or additional factors may underlie this effect. Therefore without a non-target dsRNA use, it is impossible to conclude that dsRNApvy can hinder the virus accumulation.
It seems to us that this is the main comment and our response to it may clarify many of the remaining uncertainties raised. It is common practise amongst researchers to include different non-target dsRNAs alongside target dsRNAs as controls. We included such controls in our experiments and found that only the dsRNApvy had highly nucleotide sequence specific activity. We did not include these data in the original text, given that various other reports demonstrated specific antiviral effects of the externally applied dsRNA. We have now included these control data, whereby dsRNA designed to the sequence of potato virus S genome was used to support our suggestion about specific action of dsRNApvy (Figure 1, lines 163-164).
Antiviral effects of dsRNA have been studied. However, based on current literature, it is impossible to claim that it is an RNAi-based (or RNAi-like) mechanism. The authors have shown that the effect is sequence-specific but RNAi mechanism has two fundamental features. (1) A cleavage product of the target, which can be detected by 5’RACE, 3’RACE or a Northern blot, etc. (2) sRNA signature, which leads to transitive RNAs, or a direct hint of translational repression. As the authors mention, it is true that there are many articles, where RNAi is wrongfully implicated as the mechanism. This article is no exception to that. Without showing any signs of RNAi, it is factually wrong to claim that the mechanism is RNAi-based in the title and in the text.
In the second figure, the authors performed the sRNA seq. experiment. In the second column, sRNA composition of the systemic (non-sprayed) leaves were depicted. In only 2/4 samples, they detect the sRNAs mapping to the dsRNApvy. The standard deviation of the dsRNApvy reads indicates that it is very likely to be a contamination (due to while spreading the dsRNA with a glove) rather than a physiological movement of the dsRNApvy to the systemic parts. The claim of mobile exoRNAs require further contamination controls. For example, non-sprayed leaves are collected, right after the spraying experiment (which excludes systemic movement) or RNase treatment of the contaminating RNA on the leaf surface. In addition, it is also clear that there is no reverse correlation between the read counts of dsRNApvy and PVY reads. If dsRNA really blocks the PVY -as stated in Figure 1- then the 40k dsRNA reads in Rep3 and Rep4, would have given the highest PVY reads, which is not the case (the highest PVY reads are in Rep2).
Indeed, in only 2/4 samples we detected dsRNApvy in non-treated leaves. We cannot exclude completely that this is due to contamination, however this seems to us much less likely than physiological movement in planta, because the “non-treated” leaves we used for analysis did not exist at the time of treatment but rather emerged at a later stage (there were no leaves to collect and analyse right after treatment). We have now included these clarifications in the text, and also added a statement mentioning that we cannot completely exclude contamination (lines 217-218).
The authors gave a legitimate answer to this point.
With regards to the second part of the comment, diagrams in Figure 2 represent read coverage maps for treatments with dsRNApvy only (without PVY) just to confirm that they exclusively matched the PVY genome region, which was used for dsRNA design and construction. PVY (PVY only without dsRNA) maps were obtained from separate samples only as controls and should not be compared with dsRNA maps in terms of PVY suppression (lines 197 -202).
The authors gave a legitimate answer to this point.
In Figure 3, the authors present the non-canonical sRNAs 18-30nt as a new kind of processed RNAs. However, they were previously shown in Uslu et al 2020 and Nityagoxsky et al 2022. It can simply be the degradation product of the dsRNA, which remains on the leaves, rather than any “non-canonically” processed RNAs.
The main point is that the modified Figure 1 clearly shows that an anti-PVY effect is triggered specifically via externally applied dsRNApvy (but not with non-specific dsRNA), which in accordance with previous reports, suggesting that dsRNAs induces PVY suppression through a mechanism similar to RNAi. Moreover, we clearly show that dsRNA is processed into 18-30 nt sRNAs by a yet unknown mechanism, and therefore we are functionally and mechanistically connecting these two events. When we described 18-30 nt sRNAs in the original version, we referred to other previous works: Tabein et al. [24] and Nityagovsky et al. [22]. Unfortunately, Uslu et al. (2020) was missed. In the case of Nityagovsky et al. [22] and Uslu et al. (2020), the authors also showed formation of 18-30 nt sRNA ladders from dsRNA. However, the results were contradictory: while Nityagovsky et al. [22] showed silencing effects on two endogenic/transgenic plant genes including GFP, Uslu et al. (2020) did not detect any effect on GFP expression. Tabein et al. [24] and Rego-Machado et al. [25] also showed that formation of ladder-like sRNA size distribution in N. benthamiana and tomato plants correlated with virus suppression. Reasons for these discrepancies are not clear. Our work also confirms that externally applied dsRNA induces in potato plants the formation of ladder-like sRNAs species and also a sequence-specific virus suppression, suggesting that that these two processes are mechanistically interlinked. The text has been modified accordingly (lines 246 – 257).
The authors gave a legitimate answer to this point.
In Figure 5C, the authors claim that there is the 5’ nucleotide bias -regardless of the size of the sRNA. However, they have not normalized it to the base-pair distribution of dsRNApvy. I could not find the Gene Bank accession number OR545670) but I used another accession numbers to check the PVY replicase composition: It is 30-35% A, 28-32%U -very similar to the 5’-nucleotide distribution given by in Figure 5C. Therefore, it is also an indirect proof of simple degradation, rather than any processing of the exo dsRNA.
I believe that this comment may have arisen due to a misunderstanding caused by incorrectly used phrases in our text. What we intended to state is: Non-canonical antisense (18 nt – 30 nt) sRNA species detected in dsRNApvy treated plants showed significant variations in patterns of 5’ nucleotide distribution depending on the length of the sRNA species. The distribution pattern of nucleotides at the 5’-ends of sRNAs does not generally correlate with the ratio between nucleotides in the antisense fragment of dsRNApvy. For example, although in some of the cases (e.g. 21 nt and 22 nt fragments) which were mentioned by the reviewer, distribution between A and U actually resemble A:U ratio in the dsRNApvy fragment, while C/G proportions at the 5’ ends of the sRNAs (~ 1:3) is quite different from that of the whole dsRNA fragment (~ 1:1). With some other sRNA species, patterns of 5’-terminal nucleotide distribution are even much less correlated with the A/U/C/G ratio in the dsRNA fragment. For instance, with 27 nt and 28 nt species, “A” significantly prevails over all other nucleotides (reaching ~ 50% in treated leaves); in untreated leaves, 20 nt and 23 nt species do not contain C at the 5’-end at all; 28 nt species contains neither C, nor G at the 5’ end. These data so far do not propose any sequence-selective recognition of nucleotides for processing/decay of externally applied dsRNAs. The mechanism(s) underpinning this process requires further investigation. Interestingly, any specificity (bias) of DCL cleavage of dsRNAs is also unknown. However, despite the variable distribution of nucleotides at the 5’-ends of the non-canonical sRNA, they still have potential affinity (sufficient number of A, U or C) to the same repertoire of AGO proteins (AGO1, AGO2 and AGO5) as PVY-induced siRNAs. At the same time, it is unclear if most sRNAs found in dsRNApvy treated plants, which are actually non-canonical sRNAs (with atypical size) are able to form RISCs with AGO proteins”. The text and Figure 5 have been modified accordingly and Supplementary Figure S2 has been added (lines 314 -333).
The authors clarified this point a lot better in the new version.
I would like to point out that, ‘C/G proportion at the 5’ ends of the RNAs(1:3) is “quite” different from … (1:1)” is a statement, far from a statistically meaningful claim. The same statistical analysis also lacks for 27nt, and 28nt species to show the confidence that “50% A” significantly prevails over U/C/G. For example, if A ratio was 40%, is it still significant a prevail over U/C/G ? Although the authors openly indicated that “These data so far do not propose any sequence-selective recognition of nucleotides for processing/decay of externally applied dsRNAs”, it is still important to make the claims statistically meaningful.
One suggestion is that the authors use TrAP-R instead of simple sRNA sequencing to show that the “non-canonical” sRNAs are associated with AGO proteins.
We would like to thank the reviewer for providing us with this interesting idea. We will definitely follow their advice in the future along with other approaches, such as the use of dcl and ago mutants; which will be invaluable to expand upon our good experimental system in potato and prepare a toolbox to assess the mechanistic aspects of the specific and broad spectrum DCLs, AGOs and RdRPs in potato crops.
The authors gave a legitimate answer to this point.
To sum up, the authors used sRNA-seq to assess the mechanism of exo-RNA mediated virus resistance. However, the findings are not new (18-30nt non-canonical RNAs) and they have not added any special analysis to show that these RNAs are not simple RNA degradation products on the leaf surface /or systemic leaf contaminations but they are physiological. It is impossible to reach their conclusions, simply by the given data. Rather, the 5’nucleotide analysis supports that the sRNAs were simply degradation products limited to the leaf surface.
In conclusion, we would like to thank the reviewer once again for valuable suggestions which as I hope significantly helped us to improve the paper. We have now addressed all of the comments, and our responses are listed above. We have also added many new citations, including those mentioned by the reviewer.
The authors improved the paper substantially. They share very valuable and relevant data with the scientific community. However, claiming that the mechanism of antiviral activity of dsRNA (simply based on sequence-specific activity) is RNAi-based is wrong and misleading. Therefore, I strongly encourage the authors to experimentally show proof of the principle of target RNA processing upon dsRNA or rephrase their claim about RNAi-based mechanism.
Comments on the Quality of English LanguageMinor grammatical corrections will be adequate.
Author Response
Most of the issues that I highlighted have been addressed and the strong claims of the paper have been rightfully toned down. The report and the data presented by the authors are valuable and it is worth publishing them. However, the following concepts should be rephrased to avoid any misunderstanding. These points are indicated below in green (and italic).
We are thankful to the reviewer who, we hope, helped us to improve the manuscript.
Antiviral effects of dsRNA have been studied. However, based on current literature, it is impossible to claim that it is an RNAi-based (or RNAi-like) mechanism. The authors have shown that the effect is sequence-specific but RNAi mechanism has two fundamental features. (1) A cleavage product of the target, which can be detected by 5’RACE, 3’RACE or a Northern blot, etc. (2) sRNA signature, which leads to transitive RNAs, or a direct hint of translational repression. As the authors mention, it is true that there are many articles, where RNAi is wrongfully implicated as the mechanism. This article is no exception to that. Without showing any signs of RNAi, it is factually wrong to claim that the mechanism is RNAi-based in the title and in the text.
We do agree with the reviewer, as we have also seen contradiction in our paper: from one hand we did not find any evidence of the RNAi, but from the other hand we termed the process RNAi or RNAi-like. We have now removed the claim about RNAi-based mechanism from the title as well as throughout the whole text accordingly, (see track changes; e.g. line 2; lines 13-15; lines, 27, 29, 134, 152, 155, 396, 513-514, 588, 602, 624 and all others.
The authors clarified this point a lot better in the new version.
I would like to point out that, ‘C/G proportion at the 5’ ends of the RNAs(1:3) is “quite” different from … (1:1)” is a statement, far from a statistically meaningful claim. The same statistical analysis also lacks for 27nt, and 28nt species to show the confidence that “50% A” significantly prevails over U/C/G. For example, if A ratio was 40%, is it still significant a prevail over U/C/G ? Although the authors openly indicated that “These data so far do not propose any sequence-selective recognition of nucleotides for processing/decay of externally applied dsRNAs”, it is still important to make the claims statistically meaningful.
We have dramatically reduced description and toned down interpretation of the results related to 5’-terminal ends of sRNAs leaving just the statement that "the non-canonical (18 nt – 30 nt) sRNAs species detected in dsRNApvy treated plants still have potential affinity (sufficient number of A, U or C at 5’-end) to the same repertoire of AGO proteins (AGO1, AGO2 and AGO5) as PVY-induced siRNAs (Figure 5; Supplementary Figure S2). At the same time, it is unclear if any of sRNAs found in dsRNApvy treated plants, which are actually non-canonical sRNAs (with atypical size) are able to form RISCs with AGO proteins." In principle, these data are not important for the paper, but reason we remained them is just to make these results (as well as raw data deposited at NCBI or Supplementary) available to the community.
The authors improved the paper substantially. They share very valuable and relevant data with the scientific community. However, claiming that the mechanism of antiviral activity of dsRNA (simply based on sequence-specific activity) is RNAi-based is wrong and misleading. Therefore, I strongly encourage the authors to experimentally show proof of the principle of target RNA processing upon dsRNA or rephrase their claim about RNAi-based mechanism.
This has now been completely re-phrased as suggested by reviewer (as indicated above). Thanks again.
Reviewer 2 Report
Comments and Suggestions for Authors
The manuscript is quite improved and can be accepted in its current form.
Author Response
We would like to thank the reviewer for positive comments
Round 3
Reviewer 1 Report
Comments and Suggestions for Authors
The final version of the manuscript is going to be interesting to a wide range of audience. The data presentation is clear, and the interpretation of the data is well balanced. Well done!
Comments on the Quality of English LanguageThere are minor mistakes about the use of 'the' and some minor grammatical errors. However, the text is clear and well written.